# Understanding the Interaction of Adversarial Training with Noisy Labels

## Abstract

*Noisy labels* (NL) and *adversarial examples* both undermine trained models, but interestingly they have hitherto been studied *independently*. A recent *adversarial training* (AT) study showed that the number of *projected gradient descent (PGD) steps* to successfully attack a point (i.e., find an adversarial example in its proximity) is an effective measure of the robustness of this point. Given that natural data are clean, this measure reveals an intrinsic geometric property—*how far a point is from its nearest class boundary*. Based on this breakthrough, in this paper, we figure out how AT would interact with NL. Firstly, we find if a point is *too close* to its noisy-class boundary (e.g., *one step* is enough to attack it), this point is likely to be *mislabeled*, which suggests to adopt *the number of PGD steps* as a new criterion for *sample selection* to correct NL. Secondly, we confirm that AT with strong smoothing effects *suffers less from NL* (without NL corrections) than *standard training*, which suggests that AT itself is an NL correction. Hence, AT with NL is helpful for improving even the natural accuracy, which again illustrates the superiority of AT as a *general-purpose* robust learning criterion.

## 1 Introduction

In practice, the process of data labeling is usually noisy. Thus, it seems inevitable to learn with noisy labels (Natarajan et al., 2013). To combat noisy labels, researchers have designed robust label-noise learning methods, such as sample selection (Jiang et al., 2018) and loss/label correction (Patrini et al., 2017; Nguyen et al., 2019). Meanwhile, safety-critical areas (e.g., medicine and finance) require deep neural networks to be robust against adversarial examples (Szegedy et al., 2014; Nguyen et al., 2015). To combat adversarial examples, adversarial training methods empirically generate adversarial data on the fly for updating the model (Madry et al., 2018; Zhang et al., 2019a).

An interesting fact is that, the research community is exploring label-noise learning and adversarial training *independently*. For example, Ding et al. (2020) and Zhang et al. (2021b) demonstrated that *the non-robust data that are close to the nearest class boundary are easy to be attacked*: their adversarial variants easily cross over the decision boundary. To fine-tune the decision boundaries for adversarial robustness, Ding et al. (2020) adaptively optimized small margins for non-robust data, while Zhang et al. (2021b) gave more weights on them. However, both methods in adversarial training explored the adversarial robustness with an implicit assumption that *data have clean labels*. Obviously, it is not realistic in practice. To this end, we figure out the interaction of adversarial training with noisy labels.

We discover that when noisy labels occur in adversarial training (the right panel of Figure 1), *incorrect data (square points) are more likely to be non-robust* (i.e., the predicted labels of their adversarial variants disagree with the given labels). Specifically, Figure 1 compares the difference between standard training (ST (Zhang et al., 2017)) and adversarial training (AT (Madry et al., 2018)) with noisy labels. Commonly, a small number of incorrect data (square points) are surrounded by a large number of correct data (round points). In ST, deep networks shape two small clusters (the left panel of Figure 1) around the two incorrect data due to memorization effects (Zhang et al., 2017). In contrast, AT has strong *smoothing effects*, i.e., smoothing out the small clusters around incorrect data and letting incorrect data alone (the right panel of Figure 1).

To explain the above phenomenon in AT, we believe that the adversarial counterparts generated by (majority) correct data can help to smooth the local neighborhoods of correct data, which encourages

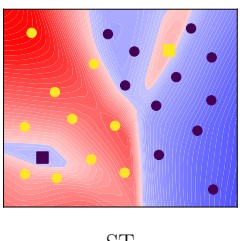
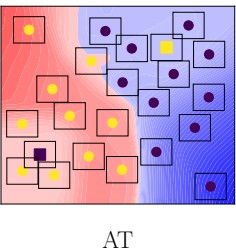

ST                    AT

Figure 1: The results of ST and AT on a binary dataset with noisy labels. Dots denote correct data; squares denote incorrect data. The color gradient represents the prediction confidence: the deeper color represents the higher prediction confidence. **Left panel**: A deep network shapes two small clusters (red and blue ones in cross-over areas) around two incorrect data due to memorization effects in ST. **Right panel**: These clusters have been smoothed out in AT. Boxes represent the unit-norm ball of AT.

Figure 2: The average entropy of models trained by ST and AT. This value is calculated on 100 points in each neighborhood of incorrect data, using *CIFAR-10* with symmetric-flipping noise. Both *solid* and *dashed* lines represent ST and AT, respectively. Note that ST learns incorrect data more deterministically than AT.

deep networks to be locally constant within the neighborhood (Papernot et al., 2016). Therefore, in AT, it becomes difficult for deep networks to form small but separated clusters around each incorrect data. Consequently, these incorrect data are non-robust, which echos the parallel findings that robust training avoids memorization of label noise (Sanyal et al., 2021).

Furthermore, we make quantitative comparisons between ST and AT in the presence of label noise. Zhang et al. (2017) showed that ST indeed overfits noisy labels, which definitely degrade the generalization performance of deep networks. From Figure 3, it can be seen that the training accuracy of deep networks on incorrect data is obviously lower than that on correct data in AT. Nonetheless, the performance gap totally disappeared in ST. Therefore, compared to ST, AT can always distinguish correct data and incorrect data. Observing Figure 4, the test accuracy of deep networks first increases then decreases in ST. Nonetheless, such a trend has been largely alleviated or totally eliminated in AT. Therefore, AT can mitigate negative effects of noisy labels, since the *smoothing effects* of AT can prevent memorizing such incorrect data.

Moreover, under noisy labels, we realize that AT provides a new measure—*how difficult it is to attack data to generate adversarial variants whose predictive labels are different from the given labels*—which can distinguish correct/incorrect data (Figures 7(a) and 7(b)) and typical/rare data (Figure 8) well. This new measure can be approximately realized by the number of projected gradient descent (PGD) steps (Madry et al., 2018), i.e., how many PGD iterations we need to generate misclassified adversarial variants. Compared with the commonly used measure, i.e., the *loss value* (Jiang et al., 2018; Han et al., 2018), we find that the *number of PGD steps* could be an alternative or even better measure in AT (Figures 7(a) and 7(b)). In addition, we discover that this new measure can easily pick up rare (atypical) data among typical data (Figure 8), where modern datasets often follow long-tailed distributions (Feldman & Zhang, 2020).

**Main contributions.** To sum up, our contributions can be summarized in three aspects as follows.

1. We explore the in-depth interaction of AT with noisy labels. Namely, we take a closer look at the smoothing effects of AT under label noise (Section 3). Subsequently, we conduct quantitative comparisons: compared with ST, AT can always distinguish correct and incorrect data and mitigate negative effects of label noise (Section 4).

2. We realize that AT naturally provides a new measure called *the number of PGD steps*, i.e., how many PGD iterations are needed to generate misclassified adversarial examples. Such a new measure can clearly differentiate the correct/incorrect data and typical/rare data (Section 5).

3. We provide two simple examples of the applications of our new measure: a) we develop a robust annotator, which can robustly annotate unlabeled (U) data considering that U data could be adversarially perturbed (Section 6.1); b) our new measure could be an alternative to the predictive probability for providing the confidence of annotated labels (Section 6.2).

## 2 BACKGROUND AND RELATED WORK

**Adversarial training (AT).** As one of the primary defenses against adversarial examples (Goodfellow et al., 2015; Carlini & Wagner, 2017; Athalye et al., 2018), AT has been widely studied to improve the adversarial robustness of deep neural networks (DNNs) (Cai et al., 2018; Wang et al., 2020b;a; Jiang et al., 2020b; Wu et al., 2020; Chen et al., 2020; Bai et al., 2021; Chen et al., 2021; Tian et al., 2021). The key objective of AT is to minimize the training loss on the adversarial variants of training data. We review the details of AT (Madry et al., 2018) used in this paper.

Let $(\mathcal{X}, d_\infty)$ denote the input feature space $\mathcal{X}$ with the infinity distance metric $d_{\inf}(x, x') = \|x - x'\|_\infty$, and $\mathcal{B}_\epsilon[x] = \{x' \in \mathcal{X} \mid d_{\inf}(x, x') \leq \epsilon\}$ be the closed ball of radius $\epsilon > 0$ centered at $x$ in $\mathcal{X}$. $S = \{(x_i, y_i)\}_{i=1}^n$ is a dataset and $(x_i, y_i)$ are i.i.d. from an underlying distribution, where $x_i \in \mathcal{X}$, $y_i \in \mathcal{Y} = \{0, 1, \ldots, C-1\}$, and $C$ denotes the number of classes. The objective function of AT is

$$\min_{f_\theta \in \mathcal{F}} \frac{1}{n} \sum_{i=1}^n \ell(f_\theta(\tilde{x}_i), y_i), \tag{1}$$

where $\tilde{x}_i$ is an adversarial variant of input data $x_i$ within the $\epsilon$-ball centered at $x$ and $f_\theta(\cdot) : \mathcal{X} \to \mathbb{R}^C$ is a score function. $\ell : \mathbb{R}^C \times \mathcal{Y} \to \mathbb{R}$ is a loss function which is a composition of a base loss $\ell_B : \Delta^{C-1} \times \mathcal{Y} \to \mathbb{R}$ (e.g., the cross-entropy loss) and an inverse link function $\ell_L : \mathbb{R}^C \to \Delta^{C-1}$ (e.g., the soft-max activation), in which $\Delta^{C-1}$ is the corresponding probability simplex—in other words, $\ell(f_\theta(\cdot), y) = \ell_B(\ell_L(f_\theta(\cdot)), y)$.

To generate the adversarial variants $\tilde{x}$ for natural data $x$, AT employs the PGD method (Madry et al., 2018). Given a starting point $x^{(0)} \in \mathcal{X}$ and step size $\alpha > 0$, PGD works as follows:

$$x^{(t+1)} = \Pi_{\mathcal{B}[x^{(0)}]}\big(x^{(t)} + \alpha \operatorname{sign}(\nabla_{x^{(t)}} \ell(f_\theta(x^{(t)}), y))\big), \tag{2}$$

until a certain stopping criterion is satisfied. In the above equation, $t \in \mathbb{N}$, $\ell$ is the loss function, $x^{(0)}$ refers to natural data or natural data perturbed by a small Gaussian or uniform random noise, $y$ is the corresponding label for natural data $x$, $x^{(t)}$ is an adversarial data point at step $t$, and $\Pi_{\mathcal{B}_\epsilon[x_0]}(\cdot)$ is the projection function that projects the adversarial data back into the $\epsilon$-ball centered at $x^{(0)}$ if necessary.

It is common to use PGD to generate adversarial variants $\tilde{x}$ in AT methods (Wang et al., 2019; Zhang et al., 2020). Recently, Zhang et al. (2021b) explored adversarial robustness by giving more weights on the non-robust data with the assumption that all labels are correct. Specifically, the non-robust data are geometrically close to the class boundaries, which can easily go across the class boundaries by a small perturbation. To approximate the distance between the data and the class boundaries, they proposed the geometry-aware projected gradient descent (GA-PGD) to calculate the *geometry value* $\kappa$, which is the least number of iterations that PGD needs to find misclassified adversarial variants of input data. In this paper, we utilize the geometry value $\kappa$ to represent our proposed measure (i.e., the *number of PGD steps*); we further explore its applications such as selecting correct/incorrect and typical/rare data (Section 5), assisting to develop a robust annotator (Section 6.1) and providing the annotation confidence (Section 6.2).

**Label-noise learning.** We consider a training set with $\mathcal{X} = (x_1, \ldots, x_N)$ and its associated labels $\mathcal{Y} = (y_1, \ldots, y_N)$, where $y_i \in \mathcal{Y}$ is the one-hot label for the instance $x_i$ and $(x_i, y_i)$ are drawn i.i.d. from some unknown distribution. In the setting of label noise, we observe noisy labels $\widetilde{\mathcal{Y}} = (\tilde{y}_1, \ldots, \tilde{y}_N)$ where $\tilde{y}_i \in \widetilde{\mathcal{Y}}$ might be different from the corresponding ground-truth label $y_i \in \mathcal{Y}$. In this paper, we mainly focus on typical class-conditional noise: 1) *symmetric-flipping noise* (Van Rooyen et al., 2015), where noisy labels are corrupted at random with the uniform distribution; 2) *pair-flipping noise* (Han et al., 2018), where noisy labels are corrupted between adjacent classes that are prone to be mislabeled. Note that pair-flipping noise is an extremely hard case of asymmetric-flipping noise (Patrini et al., 2017).

To combat noisy labels, researchers have designed robust label-noise learning methods, such as sample selection (Malach & Shalev-Shwartz, 2017; Jiang et al., 2020a; Han et al., 2020a), loss correction (Han et al., 2020b; Liu & Guo, 2020), and label correction (Wang et al., 2018). Among them, sample selection is emerging due to its simplicity. The key idea of sample selection is to back-propagate clean samples (regarded as correct data) during training. Since DNNs learn simple

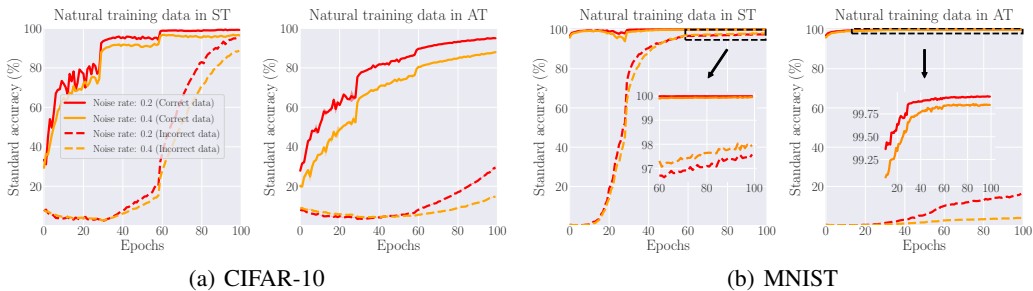

(a) CIFAR-10         (b) MNIST

Figure 3: The standard accuracy of ST and AT on correct/incorrect training data using *CIFAR-10* and *MNIST* with symmetric-flipping noise. Solid lines denote the accuracy of correct training data, while dashed lines correspond to that of incorrect training data. Compared with ST, there is a large performance gap in the standard accuracy of correct/incorrect training data in AT.

patterns first (Zhang et al., 2017; Arpit et al., 2017), the *loss value* is used as a general criterion for selecting clean samples (Han et al., 2018; Yao et al., 2020). Specifically, the data with small-loss values are considered as clean samples, which are used to update the model. In contrast, the data with large-loss values are considered as noisy samples, which should be discarded or utilized in another way (Han et al., 2020a). Note that this paper proposes a new criterion—*the number of PGD steps*—for sample selection (Section 5).

**Co-existence of adversarial examples and noisy labels (NL).** Some work has previously studied adversarial examples and noisy labels (NL) jointly. Alayrac et al. (2019) empirically showed that NL negatively affects AT's performance, but AT's robust accuracy apparently suffers less than AT's natural accuracy. Sanyal et al. (2021) empirically found that AT can avoid the memorization of NL. Damodaran et al. (2019) proposed to use Wasserstein adversarial regularization to combat NL for benefiting ST's generalization. In contrast, we advocate the AT's smoothing effect by making quantitative comparisons between AT and ST under NL. Specifically, ST has a memorization effect that gradually memorizes NL and degrades generalization in the end, while AT has a smoothing effect that avoids the memorization of NL and combats NL for benefiting generalization. In addition, we propose a new measure called the number of PGD steps for sample selection, which can differentiate the correct/incorrect data and typical/rare data, and we provide two exemplar applications for our new measure. Furthermore, in Appendix A, we provide extensive comparisons between our study and the existing literature.

## 3    Smoothing Effects of Adversarial Training

In this section, we take a closer look at the smoothing effects of AT with NL. At a high level, we conduct experiments on a synthetic dataset with incorrect labels, which explicitly show the smoothing effects of AT (Figure 1). We then use a real-world dataset, *CIFAR-10* (Krizhevsky, 2009), with incorrect labels, which further validates the smoothing effects of AT (Figure 2). As a key result, we find that AT can smooth out the small clusters around incorrect data (the right panel of Figure 1), which leads to incorrect data being non-robust in AT, i.e., they can be easily attacked to flip labels. The setup and more results can be found in Appendix B.

In detail, we empirically confirmed that DNNs can memorize random noise in ST (the left panel of Figure 1), which has been found in previous works (Zhang et al., 2017; Arpit et al., 2017). However, a recent study (Sanyal et al., 2021) claimed that AT can avoid the memorization of incorrect data through *analyzing model predictions*. Going beyond their analysis, we further investigated AT with noisy labels and provided an in-depth explanation, namely *smoothing effects*. Specifically, AT prevents incorrect data from forming small clusters, which should be the primary reason for avoiding the memorization of incorrect data.

To justify our smoothing effect, we performed a series of comparison experiments using ST and AT on a synthetic dataset with incorrect labels. In Figure 1, the model trained by ST can overfit the incorrect data (yellow and black squares), and thus have incorrect predictions (red and blue clusters in cross-over areas) around incorrect data. While in AT (with smoothing effects), such clusters have obviously disappeared. The reason is due to the smoothing effects from the adversarial variants

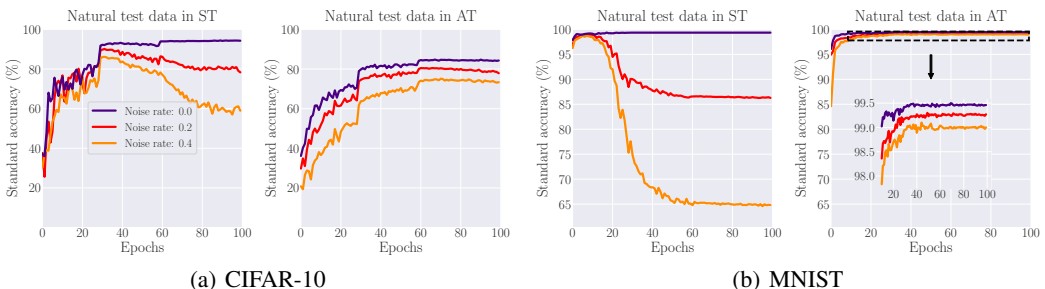

(a) CIFAR-10             (b) MNIST

Figure 4: The standard accuracy of ST and AT on natural test data using *CIFAR-10* and *MNIST* with symmetric-flipping noise for training. Note that the larger noise rate causes the test accuracy of ST dropping more seriously due to memorization effects in deep learning, while AT alleviates such negative effects.

generated from correct data. Namely, the number of correct data is larger than that of incorrect data. Thus, it is difficult for incorrect data to smooth their neighborhood.

Further, we calculated the entropy values of the model predictions on the *CIFAR-10* dataset, which aims to validate the smoothing effect in practice. Specifically, we randomly selected 100 points in each neighborhood (within a small $\epsilon$-ball) of the incorrect data and calculated their average entropy values in training (Figure 2). As a measure of uncertainty (Dai & Chen, 2012), the entropy value was calculated by the following formula:

$$H(\mathcal{Y}|\mathcal{X}) = -\sum_{x \in \mathcal{X}} \sum_{y \in \mathcal{Y}} p(x, y) \cdot \log p(y|x). \tag{3}$$

The smaller value represents the higher certainty of model prediction (and vice versa), which indicates that the model learns the data more deterministically. Thus, the higher certainty leads to the higher possibility of incorrect data forming small clusters in their neighborhoods.

We compared the entropy values of ST and AT. During the training process, under the same noise rate, the entropy value of AT is always higher than that of ST. After epoch 60, the entropy value of ST drops very fast, while that of AT remains high. It clearly shows that smoothing effects in AT prevent the model from learning incorrect data with their neighborhoods deterministically, which further confirms that it is harder for incorrect data to form small clusters. By observing Figures 1 and 2, we confirmed that it is difficult for incorrect data in AT to form small clusters due to the smoothing effects from the adversarial variants of correct data.

## 4 KNOCK-ON EFFECTS OF ADVERSARIAL TRAINING

In this section, we explore knock-on effects of AT comprehensively. We show the quantitative differences between ST and AT with noisy labels. First, in terms of training accuracy, we show that correct/incorrect data can be always distinguishable in AT (Figure 3). Second, in terms of test accuracy, we demonstrate that AT alleviates negative effects of incorrect data and then improves the model generalization (Figure 4). Note that we display the experimental results on the *CIFAR-10* and *MNIST* datasets (LeCun et al., 1998) with symmetric-flipping noise in this section. More results (e.g., pair-flipping noise, the *CIFAR-100* dataset, the loss values, and different networks) can be found in Appendix C.

### 4.1 DISTINGUISHABLE CORRECT/INCORRECT DATA

In Figure 3, we plotted the standard accuracy of natural *training* data in ST and AT. In the early stage of training, there is a clear performance gap between the standard accuracy of ST on correct/incorrect training data. However, after 60 epochs, the standard accuracy of ST on incorrect training data increases rapidly, while that of AT on incorrect training data rises relatively slowly. When the training comes to epoch 100, the standard accuracy of ST on correct/incorrect training data are merged together. Nonetheless, there is still a large performance gap in AT. Compared to the results on *CIFAR-10*, such a gap is more obvious on *MNIST*.

### 4.2 ALLEVIATION OF MEMORIZATION EFFECTS

In Figure 4, we plotted the standard accuracy of natural *test* data in ST and AT. It shows that AT can alleviate the negative effects of label noise and then improve the model generalization. Specifically, the larger noise rate causes the test accuracy of ST to drop more seriously, i.e., memorization effects (Arpit et al., 2017). However, AT reduces such negative effects. By checking the standard accuracy of natural test data, we find that there is no obvious overfitting phenomenon in AT. We also visualized the loss landscape (Li et al., 2018) of models trained by ST and AT to further substantiate the alleviation, which can be found in Appendix C.

It is worthwhile to observe the results on *MNIST*: *simply using AT can make the model obtain a performance similar to noise-free training*. However, on more complex *CIFAR-10*, incorrect data still have a certain negative impact on the model trained by AT. To reduce such an impact, a simple yet effective method is to use sample selection to filter correct/incorrect data for training (Jiang et al., 2018; Cheng et al., 2021). Therefore, it is critical to have a measure which can provide the stratification for correct/incorrect data. Normally, the *loss value* can be a good candidate in ST. However, in AT, we can find a better measure such as the *number of PGD steps* (i.e., geometry value $\kappa$). Since the smoothing effects in AT can make incorrect data be non-robust, the geometry value $\kappa$—how difficult it is to attack data to let them go across the decision boundary—could be naturally used as a measure for this task.

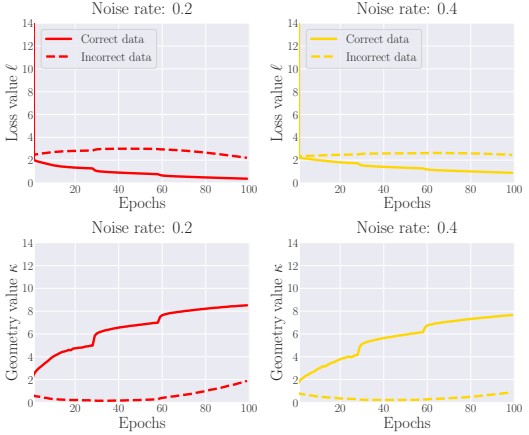
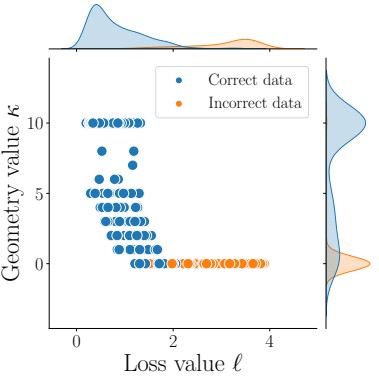

Figure 5: Comparisons of correct/incorrect data in terms of the loss value (top panel) and the geometry value $\kappa$ (bottom panel) on *CIFAR-10* with symmetric-flipping noise in AT. We calculate the mean values in each epoch. We clearly demonstrate that the value $\kappa$ has a similar trend as loss value in AT; both can be used for differentiating correct/incorrect data in AT.

Figure 6: We choose the model trained by AT using *CIFAR-10* with 20% symmetric-flipping noise. We jointly analyze the geometry value $\kappa$ and the loss value, which shows that the value $\kappa$ can provide a fine stratification on typical (i.e., larger $\kappa$)/rare (i.e., smaller $\kappa$) data.

## 5 NEW MEASURE: GEOMETRY VALUE $\kappa$

In this section, we show the geometry value $\kappa$ could be a new measure for the data stratification. First, the geometry value $\kappa$ can differentiate correct/incorrect data in AT (Figures 5, 7(a) and 7(b)). Compared with the loss value, which has been widely used in sample selection (Jiang et al., 2018; Han et al., 2018; Yu et al., 2019), we show that the geometry value $\kappa$ can have a better performance to filter incorrect data with different noise types. Second, we demonstrate that the geometry value $\kappa$ can provide a finer stratification on typical/rare data (Figures 6 and 8). We also discussed it with different configurations (e.g., PGD step number or the $\epsilon$-ball) and more results can be found in Appendix D.

### 5.1 GEOMETRY VALUE VS. LOSS VALUE

To combat NL, sample selection methods are very effective. As a common measure in sample selection, the loss value is used to filter incorrect data. For example, small-loss data can be regarded

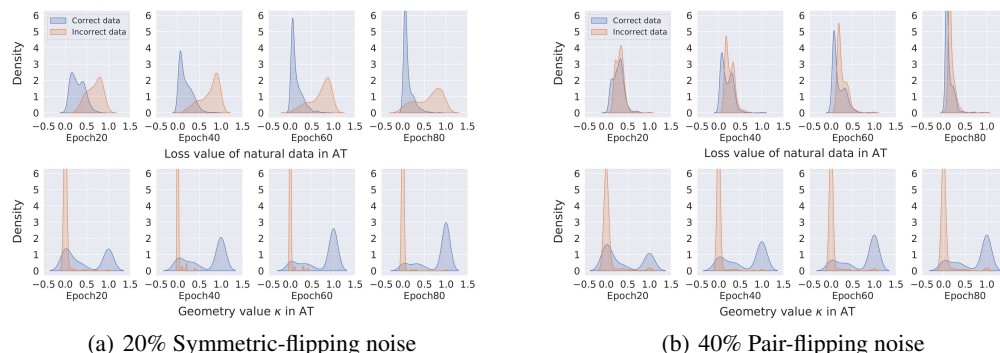

(a) 20% Symmetric-flipping noise         (b) 40% Pair-flipping noise

Figure 7: The density of AT on correct/incorrect data using *CIFAR-10* with (a) 20% symmetric-flipping noise and (b) 40% pair-flipping noise. *Top panels*: the loss value in AT. *Bottom panels*: the geometry value $\kappa$ in AT. The geometry value $\kappa$ has a better distinction on correct/incorrect data.

as "correct" data. However, there are two limitations in using the loss value as a measure. First, we need to adjust different thresholds to obtain a better selection effect, when the dataset has different noise rates and types (Yao et al., 2020). Second, for pair-flipping noise, the loss value cannot distinguish correct/incorrect data well (the top panel of Figure 7(b)).

In Figure 5, we compared the geometry value $\kappa$ and loss value of correct/incorrect data in the training process. We found that the value $\kappa$ can be used to differentiate incorrect data from correct data, since it has a similar trend to the loss value in AT. To further compare two measures in distinguishing correct/incorrect data, we plotted the density maps of two measures on the *CIFAR-10* dataset with different noise types in Figures 7(a) and 7(b). To compare the two measures in a meaningful way, we performed the min-max normalization (Tax & Duin, 2000) on both the loss value and geometric value $\kappa$, which scales the range of values in $[0, 1]$.

For symmetric-flipping noise (Figure 7(a)), although the loss value can distinguish correct data from incorrect data during the training process, the geometric value $\kappa$ has a better distinction between correct and incorrect data. Specifically, the top panels of Figure 7(a) show that there are a large number of correct/incorrect data with the same loss value, which requires a carefully designed threshold to select the correct data from incorrect data. In contrast, correct/incorrect data can be well divided using the value $\kappa$ in the bottom panels of Figure 7(a). We can easily select correct/incorrect data with high purity. More obviously, for pair-flipping noise, the loss value of correct/incorrect data overlaps in the top panels of Figure 7(b). However, the value $\kappa$ in the bottom panels of Figure 7(b) still provides a good discrimination on correct/incorrect data.

In addition, we found that the geometry value $\kappa$ can provide a fine stratification on typical/rare data. First, we jointly analyzed the value $\kappa$ and the loss value in AT (Figure 6), where we stratified correct data via $\kappa$. Secondly, by inspecting the semantic information with different $\kappa$, we found that the value $\kappa$ can represent whether the data is relatively typical or rare (Figures 8(a) and 8(b)). Moreover, we plotted a bivariate graph of the loss value and the value $\kappa$ in Figure 6. In this figure, we mainly focused on the correctly classified data (blue scattered dots), since the wrongly classified data (orange scattered dots) had been clearly discriminated by big loss values. Note that, for small-loss (correct) data, the value $\kappa$ can further subdivide such data into typical and rare types.

## 5.2 DISTINGUISHABLE TYPICAL/RARE DATA

From the *macro* perspective, the loss value can be regarded as a measure to classify correct and incorrect data (Jiang et al., 2018). Namely, small-loss data can be regarded as correct data, and vice versa. However, such stratification is a bit rough, which motivates us to seek a *micro* measure called the geometry value $\kappa$ (the number of PGD steps) in AT. To justify our findings in Figure 6, we visualized the semantic information of *CIFAR-10* (Figure 8(a)) and *MNIST* (Figure 8(b)) under different $\kappa$. We found that images with large $\kappa$ (rightmost) are prototypical and easier to recognize from the viewpoint of human perception, while images with small $\kappa$ (or $\kappa = 0$) seem to be rarer (or incorrect). These rare images have atypical semantic information, such as some strange shapes ("8" with $\kappa = 14$ in Figure 8(b)) or confusing backgrounds ("deer" with $\kappa = 2$ in Figure 8(a)). More results about the images with different $\kappa$ can be found in Appendix D.

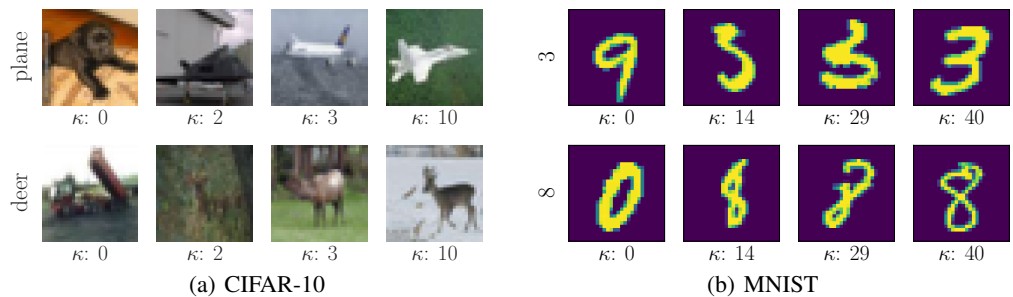

(a) CIFAR-10               (b) MNIST

Figure 8: The geometry value $\kappa$ w.r.t. images in *CIFAR-10* and *MNIST* with 20% and 10% symmetric-flipping noise. The leftmost of each subfigure is the given label (i.e., *deer* and *plane* or 3 and 8) of all images on the right. We randomly select four examples with the different $\kappa$ in each class. As the geometric value $\kappa$ increases from left ($\kappa = 0$) to right ($\kappa = 10$ or $40$), the semantic information of images is more typical and recognizable.

---

**Algorithm 1:** Robust Annotation Algorithm.

---

**Input**   :network $f_\theta$, training dataset $S = \{(x_i, y_i)\}_{i=1}^n$, learning rate $\eta$, number of epochs $T$, batch size $m$, number of batches $M$, threshold for geometry value $K$, threshold for loss value $L$.

**Output** :robust annotator $f_\theta$.

**for** epoch $= 1, \ldots, T$ **do**

    **for** mini-batch $= 1, \ldots, M$ **do**

        **Sample:** a mini-batch $\{(x_i, y_i)\}_{i=1}^m$ from $S$.

        **for** $i = 1,\ldots,m$ *(in parallel)* **do**

            **Calculate:** $\kappa_i$ and $\ell_i$ of $(x_i, y_i)$.

            **if** $\kappa_i < K$ *and* $\ell_i > L$ **then**

                **Update:** $y_i \leftarrow \arg\max_i f_\theta(x)$.

            **end**

            **Generate:** adversarial data $\tilde{x}_i$ by PGD method.

        **end**

        **Update:** $\theta \leftarrow \theta - \eta \nabla_\theta \{\ell(f_\theta(\tilde{x}_i), y_i)\}$.

    **end**

**end**

---

## 6   Applications of Geometry Value $\kappa$

In this section, we provide two applications of our new measure—the geometry value $\kappa$ (the number of PGD steps). Since the value $\kappa$ can differentiate correct/incorrect data in AT (Section 5.1), in the presence of label noise, we can use it to detect noisy labels and correct labels (Figure 9). Meanwhile, as it can have a fine stratification for typical/rare data (Section 5.2), we can provide the confidence of annotated labels according to the value $\kappa$ (Figure 10).

Regardless of ST or AT, high-quality training data are always essential for acquiring a good model (Deng et al., 2009), but the labeling process of high-quality data requires a lot of human resources. To deal with such a problem, many methods used ST to facilitate a *standard annotator* to annotate large-scale unlabeled (U) data (Carmon et al., 2019; Alayrac et al., 2019). However, this standard annotator fails when U data are adversarially manipulated.

In practice, label-noise issues widely exist in real-world training datasets, and learning with NL seems inevitable. Meanwhile, the existence of adversarial examples (Szegedy et al., 2014; Goodfellow et al., 2015) also poses a threat to annotate U data. Therefore, we design a robust annotation algorithm (Algorithm 1) to assign reliable labels for U data even in the presence of adversarial manipulations and noisy training labels (Section 6.1). Compared to human beings, the standard annotator cannot give the information whether the label assignment for U data is reliable. Nonetheless, our new measure could be an alternative to the predictive probability for providing the confidence of annotated labels (Section 6.2). The detailed experimental setups can be found in Appendix E.

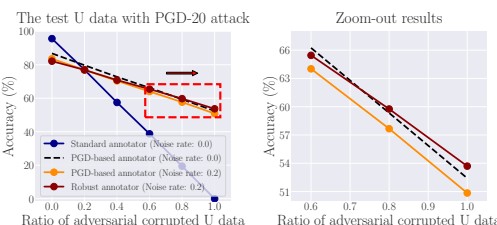 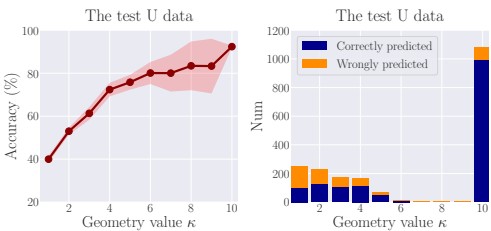

Figure 9: The accuracy of four approaches assigning correct labels to adversarial U data from *CIFAR-10*. *Left panel:* the full results. *Right panel:* the zoom-out results (without standard annotator). Our robust annotator has a satisfactory performance on assigning reliable labels.

Figure 10: The accuracy (left panel) and number (right panel) of correctly predicted U data w.r.t. the geometry value $\kappa$. We randomly select 2000 test data in *CIFAR-10* as unlabeled data. The larger $\kappa$ corresponds to the higher prediction accuracy.

## 6.1 ROBUST ANNOTATOR

We can construct a robust annotator to assign labels for U data. Here, we consider a real-world scenario, namely, existence of label noise in training data and adversarial manipulations in U data. Our robust annotator has a better labeling performance than the standard annotator, since we use the value $\kappa$ and the loss value jointly to select incorrect training data. We re-annotate high-quality pseudo labels for these incorrect data, and adversarially train on the whole data. Then, our robust annotator can reliably assign labels.

In Figure 9, we tested the accuracy of assigning correct labels to U data in the presence of adversarial manipulations. We compared four methods, namely, our robust annotator with $20\%$ symmetric-flipping noise (red line), the PGD-based annotator with $20\%$ symmetric-flipping noise (orange line), the PGD-based annotator without noise (oracle, black dashed line), and the standard annotator without noise (blue line). On normal U data (i.e., zero adversarial ratio), the standard annotator has better performance of labeling. However, when U data is subject to certain adversarial manipulations (i.e., ratio above 0.2), the labeling quality of the standard annotator decreases sharply, but that of our robust annotator still remains satisfactory. An extreme case is that, when all U data (ratio 1.0) are added to adversarial manipulations, labels assigned by the standard annotator become completely unreliable, but our labels assigned by the robust annotator are still better than the PGD-based annotator with $20\%$ symmetric-flipping noise.

## 6.2 CONFIDENCE SCORES

For a given data point, the geometry value $\kappa$ can provide a confidence score, which represents the reliability of label annotations. The measure value $\kappa$ can distinguish between typical data (correctly labeled with high probability) and rare data (wrongly labeled with high probability) in U data. In the left panel of Figure 10, we plotted the accuracy of correctly predicted data with the value $\kappa$. The larger $\kappa$ corresponds to higher prediction accuracy, which shows that the value $\kappa$ can indeed represent the reliability of label annotations. In the right panel of Figure 10, we further investigated the number of correctly predicted data with the value $\kappa$. Most of the data have the value $\kappa = 10$, which corresponds to a high prediction accuracy. Meanwhile, a small part of the data have the value $\kappa \in [0, 6]$, which corresponds to a low prediction accuracy. Since the number of data with value $\kappa \in [7, 9]$ is small, the standard deviation of the accuracy is large.

## 7 CONCLUSION

In this paper, we explored the interaction of adversarial training (AT) with noisy labels. We took a closer look at smoothing effects of AT, and further investigated positive knock-on effects of AT. As a result, AT can distinguish correct/incorrect data and alleviate memorization effects in deep networks. Since smoothing effects can make incorrect data non-robust, the geometry value $\kappa$ (i.e., the number of projected gradient descent steps) could be a new measure to differentiate correct/incorrect and typical/rare data. Moreover, we gave two applications of our new measure, i.e., the robust annotator and confidence scores. With the robust annotator, we can assign reliable labels for adversarial U data. With confidence scores, we can know the reliability of label annotations.

## 8 ETHICS STATEMENT

This paper does not raise any ethics concerns. This study does not involve any human subjects, practices to data set releases, potentially harmful insights, methodologies and applications, potential conflicts of interest and sponsorship, discrimination/bias/fairness concerns, privacy and security issues, legal compliance, and research integrity issues.

## 9 REPRODUCIBILITY STATEMENT

To ensure the reproducibility of experimental results, we will provide a link for an anonymous repository about the source codes of this paper in the discussion forums.

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

## A  DETAILED DISCUSSIONS ON THE DIFFERENCES WITH THE EXISTING STUDIES

This section discusses the difference between our work and other related studies that focused on either improving adversarial training (AT) or learning with noisy labels (NL) in ST. Our work focuses on figuring out the in-depth interaction of (canonical) adversarial training with (generalized) noisy labels.

**Improving adversarial training (AT).**    Some studies focused on improving the AT's performance by leveraging additional unlabelled (U) data (Carmon et al., 2019; Najafi et al., 2019; Alayrac et al., 2019; Zhang et al., 2019b). The main points are leveraging additional U data in AT that can achieve both higher robust accuracy and higher standard accuracy. The ablation study of (Alayrac et al., 2019) provided an analysis on the impact of symmetric-flipping noise for model robustness, which simulates the unreliable annotation of U data. By comparison, our study claims that AT itself an NL correction. We conduct experiments on the generalized setting of NL, i.e., with different kinds of label noise (e.g., symmetric-flipping noise and pair-flipping noise) and different noise rates (e.g., [0.1,0.4]).

For benefiting adversarial robustness, Zhang et al. (2021a) proposed to inject NL over the training process. They assume that the training set is noise-free, and they inject NL on the fly as AT's regularization method. By comparison, our settings and motivations are different. We assume the training set is label-noisy and find that AT can naturally mitigate the negative effect of NL in the training set. We focus on the understanding of AT's smoothing effects on NL.

**Learning with noisy labels (NL) in ST.**    Damodaran et al. (2019) focused on learning with NL in ST and designed a Wasserstein Adversarial Regularization (WAR) as a correction method. They added the WAR into standard training (ST) to combat with the NL. The authors also provided analysis for their proposed correction method. Specifically, they explained WAR as a label interpolation that uses the prediction of the adversarial data to interpolate the original label for the natural data. By comparison, we identified and illustrated the inherent reason for AT's NL correction, i.e., the smoothing effect of the AT. Due to AT's smoothing effect, the predictions of the adversarial data are more trustworthy and can be used to interpolate the noisy labels.

Kaneko et al. (2019) focused on robust label noise learning in the generation task. They proposed a robust GAN to combat with NL. Specifically, they introduced a noise transition model as an auxiliary classifier for discriminator, which is similar to the forward correction method in label noise learning. Their proposal conduct adversarial learning (AT) for generating images. By comparison, we leverage AT for defending adversarial examples and meanwhile correcting NL.

Sanyal et al. (2021) found NL in ST causes the significant adversarial vulnerability. Besides, they found NL widely exists even in some standard datasets (e.g., *MNIST* and *CIFAR-10*) and identified NL as one of the causes for adversarial vulnerability in ST. In terms of AT, they empirically found that AT can avoid the memorization of NL by conducting the experiments using symmetric-flipping noise on AT. By comparison, we figure out why AT can avoid the memorization of NL, i.e., AT's smoothing effect. Furthermore, we conduct extensive experiments across different noise types (e.g., asymmetric-flipping noise) on AT and make comprehensive comparisons with ST.

To sum up, our main point is understanding the interaction of AT with NL, which is different from the previous studies. Specifically, we have shown the AT's *smoothing effects* on NL and identified that AT itself an NL correction. Therefore, compared with ST, AT can avoid the memorization of NL and make correct (clean) data and incorrect (noise) data always distinguishable. Besides the discoveries, we proposed a new measure—*PGD steps* that can stratify correct/incorrect and typical/rare data in AT, which may provide a new perspective for sample selections.

## B  THE SMOOTHING EFFECTS OF ADVERSARIAL TRAINING

In this section, We provide the detailed setup and more results on the synthetic binary dataset and the real-world dataset (*CIFAR-10*) with noisy labels, which demonstrate the smoothing effects of adversarial training (AT).



| ST | AT (PGD-1) | AT (PGD-2) | AT (PGD-3) | AT (PGD-4) |

Figure 11: The results of standard training (ST) and adversarial training (AT) on a binary dataset with noisy labels. Dots denote correct data, while squares denote incorrect data. The color gradient represents the prediction confidence: the deeper color represents higher prediction confidence. In the leftmost panel, deep networks shapes two small clusters (red and blue ones in cross-over areas) around two incorrect data due to memorization effects in ST. As the number of PGD iterations increases, the smoothing effects in AT gradually strengthens, and two small clusters gradually shrink until they disappear in the rightmost panel. Namely, these clusters have been smoothed out in AT (PGD-4). Boxes represent the norm ball of AT.

**Experimental setup.** To construct synthetic binary dataset, we randomly generate 23 points (i.e., $(a, b)$, where $a \in (0, 1)$ and $b \in (0, 1)$) with binary labels (i.e., "0" and "1") on a two-dimensional plane. Among all data , we choose two points to assign incorrect labels. For the binary classification, we build a simple network contains 5 linear layers and 4 ReLU (Nair & Hinton, 2010) layers. We train the simple network in ST and AT using Adam with the initial learning rate=0.001 for 1000 iterations. In AT, we set the perturbation bound $\epsilon = 0.08$ and the PGD step size $\alpha = 0.02$.

**Result.** In Figure 11, We plot the classification results in the two-dimensional plane for both ST and AT. We use different PGD iterations to generate adversarial examples, which shows the smoothing process dynamically. In ST, deep network will shape two small clusters around two incorrect data due to memorization effects. While in AT, these small clusters will gradually shrink until they disappear, as the smoothing effects in AT strengthens (i.e., from PGD-1 to PGD-4).

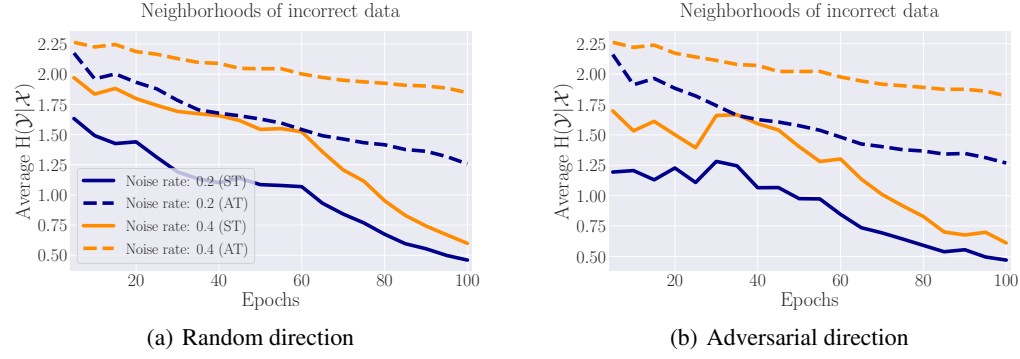

(a) Random direction           (b) Adversarial direction

Figure 12: The average entropy of models trained by ST and AT. This value is calculated on 100 points in each neighborhood of incorrect data, using *CIFAR-10* with symmetric-flipping noise. Both *solid* and *dashed* lines represent ST and AT, respectively. Note that ST learns incorrect data more deterministically than AT.

**Result.** In Figure 12, we plot the entropy values of the model predictions on the *CIFAR-10* dataset. In Figure 12(a), we randomly select 100 points in each neighborhood (within a small $\epsilon$-ball, where $\epsilon = 0.031$) of the incorrect data and calculate their average entropy values in training using the models trained by ST and AT. In Figure 12(b), we generate the adversarial variant for each incorrect data by PGD-1 attack with $\epsilon = 0.031$ and calculate the average entropy values in training. The detailed training settings can be found in Appendix C.1. On the whole, compared with ST, the entropy values in AT are always higher. It demonstrates that AT did not learn incorrect data with their neighborhoods deterministically, which confirms that the smoothing effects in AT prevent incorrect data from forming small clusters during training.

## C KNOCK-ON EFFECTS OF ADVERSARIAL TRAINING

In this section, we provide more complementary experiments and analysis for the positive knock-on effects of AT. First, we show the results of standard training and test accuracy on *CIFAR-10* and *MNIST* datasets with different noise rates and types (Appendix C.1). Second, we show the analysis of the natural data and adversarial data in AT (Appendix C.2). Third, we use different networks to investigate positive knock-on effects of AT with noisy labels (Appendix C.3). Finally, we show the results of loss value with different noise rates and types (Appendix C.4).

### C.1 TRAINING ACCURACY AND TEST ACCURACY

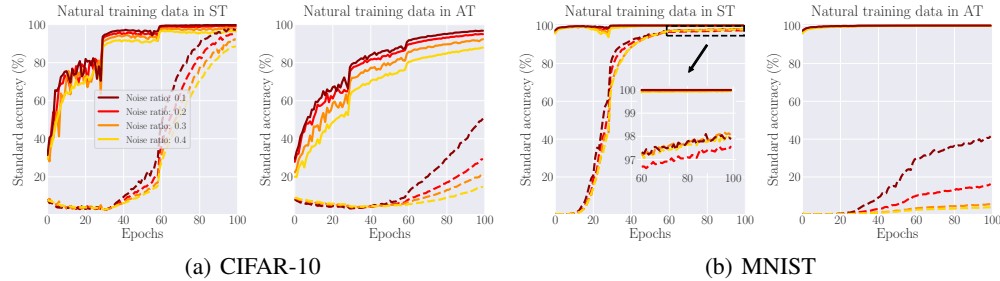

(a) CIFAR-10          (b) MNIST

Figure 13: The standard accuracy of ST and AT on correct/incorrect training data using *CIFAR-10* and *MNIST* with *symmetric-flipping* noise. Solid lines denote the accuracy of correct training data, while dashed lines correspond to that of incorrect training data. Compared with ST, there is a large performance gap in the standard accuracy of correct/incorrect training data in AT.

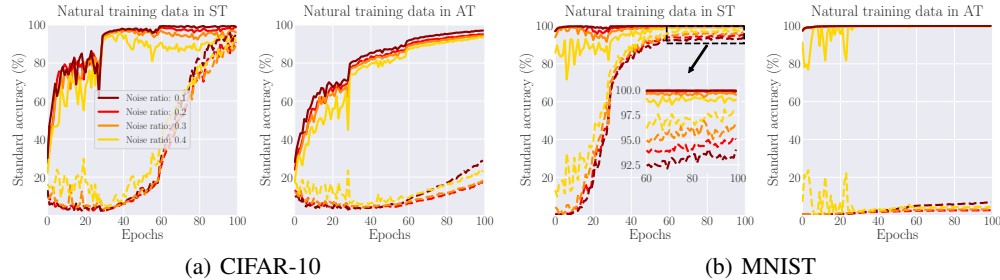

(a) CIFAR-10          (b) MNIST

Figure 14: The standard accuracy of ST and AT on correct/incorrect training data using *CIFAR-10* and *MNIST* with *pair-flipping* noise. Solid lines denote the accuracy of correct training data, while dashed lines correspond to that of incorrect training data. Compared with ST, there is a large performance gap in the standard accuracy of correct/incorrect training data in AT.

**Experimental setup.** We conduct our experiments on two datasets with different noise rates (e.g., $[0.1, 0.4]$) and different noise types (e.g., symmetric-flipping noise and pair-flipping noise). We use the method in (Han et al., 2018) to generate noisy training data. For the *CIFAR-10* dataset, we normalize it into $[0, 1]$: Each pixel is scaled by $1/255$. We perform the standard *CIFAR-10* data augmentation: a random 4 pixel crop followed by a random horizontal flip. For the *MNIST* dataset, we normalize it into $[0, 1]$. We train ResNet-18 in ST and AT using SGD with $0.9$ momentum for 100 epochs on *CIFAR-10* dataset. The initial learning rate is $0.1$ divided by 10 at Epoch 30 and 60 respectively. The weight decay=$0.0005$. For *MNIST* dataset, we use SmallCNN (Zhang et al., 2019a), and set the initial learning rate as $0.01$. The rest of the settings remain the same as training on *CIFAR-10*. In AT, we set the perturbation bound $\epsilon = 0.031$, the PGD step size $\alpha = 0.007$, and PGD step numbers $K = 10$. For standard evaluation, we obtain standard accuracy on natural training data according to correct/incorrect labels and natural test data with all correct labels. For the robust

evaluation, we obtain robust accuracy on adversarial training and adversarial test data. The adversarial test data are generated by PGD-20 attack with the same perturbation bound $\epsilon = 0.031$ and the step size $\alpha = 0.031/4$, which keeps the same as Wang et al. (2019). All PGD generation have a random start, i.e, the uniformly random perturbation of $[-\epsilon, \epsilon]$ added to the natural data before PGD iterations.

**Result 1.** In Figures 13 and 14, we plot the standard accuracy of correct/incorrect training data with different noise rates and types. On the whole, compared with ST, correct/incorrect training data can be always distinguishable in AT regardless of noise rates and types. Compared with symmetric-flipping noise, AT has a better performance on distinguishing correct/incorrect data with pair-flipping noise. Note that, under the same noise rates, the standard accuracy on incorrect training data in AT with pair-flipping noise is lower than that with symmetric-flipping noise.

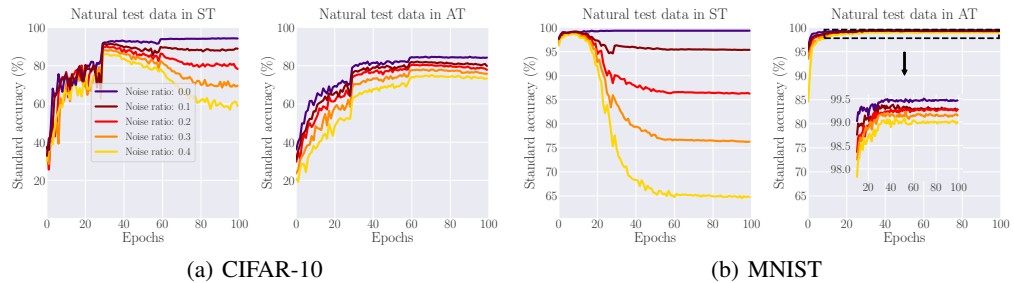

Figure 15: The standard accuracy of ST and AT on natural test data, where training data using *CIFAR-10* and *MNIST* with *symmetric-flipping* noise. Note that the larger noise rate causes the test accuracy of ST dropping more seriously due to memorization effects in deep learning, while AT alleviates such negative effects.

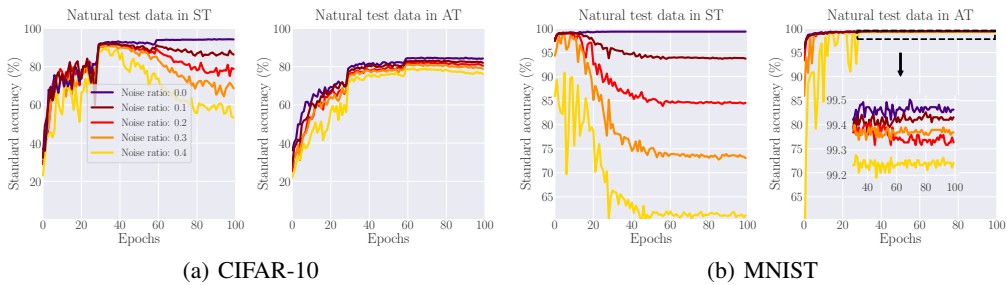

Figure 16: The standard accuracy of ST and AT on natural test data, where training data using *CIFAR-10* and *MNIST* with *pair-flipping* noise. Note that the larger noise rate causes the test accuracy of ST dropping more seriously due to memorization effects in deep learning, while AT alleviates such negative effects.

**Result 2.** In Figures 15 and 16, we plot the standard accuracy of natural test data with different noise rates and types. On the whole, AT can alleviate the negative effects of label noise due to memorization effects in deep learning. Under each noise type, as the noise rate increases, standard accuracy of ST on natural test data drops more seriously in the later stage of training (e.g., after 60 epochs in *CIFAR-10*). In AT, we only observe that the larger noise rate causes the lower standard accuracy on natural test data, while the overfitting phenomenon is not obvious. Compared with *CIFAR-10*, both symmetirc-flipping and pair-flipping noise have more serious negative effects on *MNIST*, while simply using AT can alleviate these effects to a greater extent. In Figure 17, we also visualize the loss landscape (Li et al., 2018) of models trained by ST and AT on *CIFAR-10*. Such visualization can further substantiate that AT mitigates negative effects of label noise via the lens of the model generalization. Namely, the loss landscape w.r.t. weight space of an adversarially trained model (i.e., AT) is smoother and flatter than that of a model using ST.

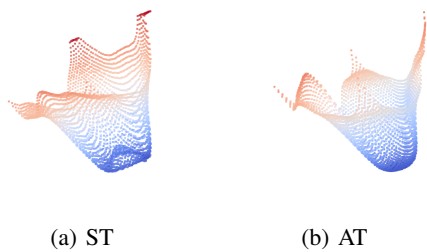

(a) ST  (b) AT

Figure 17: The loss landscape w.r.t weight space of models trained by ST and AT using *CIFAR-10* with 20% symmetric-flipping noise. The red/blue colors denote large/small values, which reflect the relative position in the loss landscape. Note that the loss landscape of a model trained by AT is smoother and flatter than that by ST, which reflects the better model generalization by AT.

We also conduct experiments on a harder dataset, i.e., *CIFAR-100*. The smoothing effect of AT is not significant as that on *CIFAR-10* or *MNIST* but can also be found on this dataset. Specifically, there is a larger performance gap in the standard accuracy of correct/incorrect training data in AT (e.g., 24% -30%) than ST (e.g., 0.15% -0.25%), and the test accuracy of ST dropping more seriously (e.g., 2% -16%) than AT (e.g., 1% -7%).

## C.2 NATURAL DATA AND ADVERSARIAL DATA

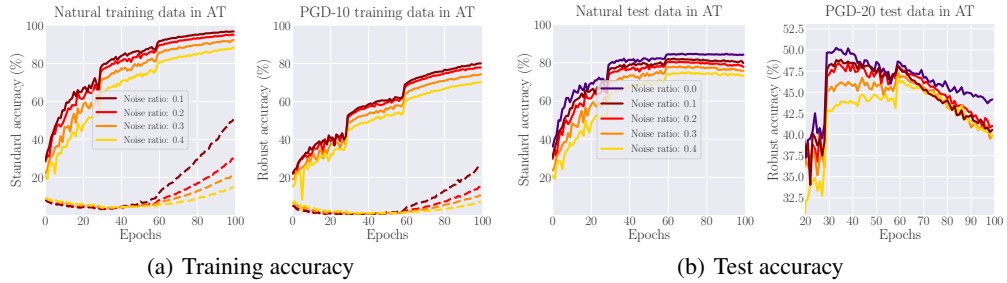

(a) Training accuracy  (b) Test accuracy

Figure 18: The standard/robust accuracy of AT on natural training data, adversarial training data (PGD-10), adversarial test data (PGD-20) using the *CIFAR-10* dataset with *symmetric-flipping* noise.

**Result.** In Figure 18, we plot the standard and robust accuracy on natural data and adversarial data (e.g., PGD-10 training data and PGD-20 test data) using *CIFAR-10* with symmetirc-flipping noise. Different from ST, each natural training data will generate a corresponding adversarial data in AT. We also check the difference in robust accuracy between correct and incorrect adversarial data during training. We found that AT can also distinguish correct/incorrect adversarial data over the whole training process. However, we find that the difference between correct and incorrect adversarial data (right panel in Figure 19(a)) is smaller than that between incorrect and correct natural data (left panel in Figure 19(a)).

## C.3 DIFFERENT NETWORKS

**Result.** In Figure 19, we plot the standard accuracy on natural training/test data using ResNet-10, ResNet-18, ResNet-26 and ResNet-34 trained by ST and AT. We conduct the experiments using *CIFAR-10* dataset with 20% symmetric-flipping noise. The training settings keep the same as Appendix C.1. We find that, using different networks, AT still has a better performance on distinguishing correct/incorrect data compared with ST and can alleviate the negative effects of label noise.

**Result.** In Figure 20, we plot the standard accuracy on natural test data using a large deep network, Wide-ResNet (e.g.,WRN-32-10 (Zagoruyko & Komodakis, 2016)), trained by ST and AT. We

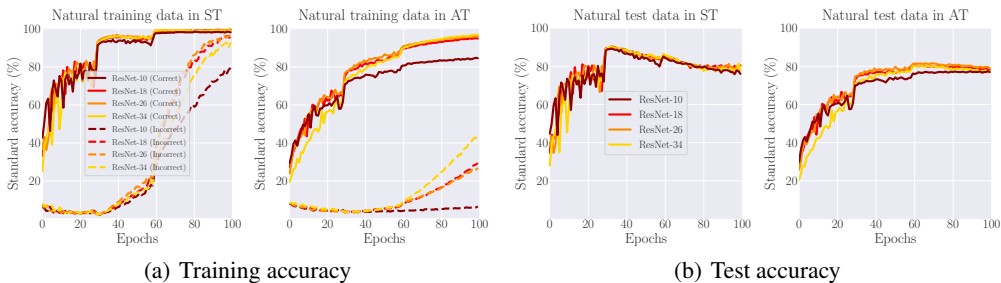

(a) Training accuracy                    (b) Test accuracy

Figure 19: The standard accuracy of AT on natural training/test data using the *CIFAR-10* dataset with 20% *symmetric-flipping* noise. We conduct the experiments using ResNet-10, ResNet-18, ResNet-26 and ResNet-34.

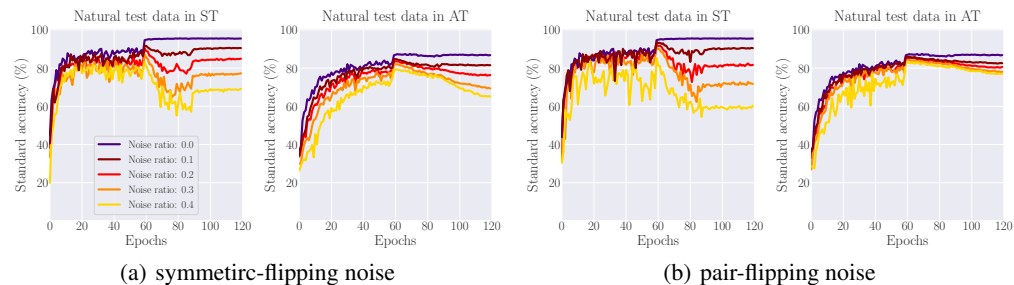

(a) symmetirc-flipping noise            (b) pair-flipping noise

Figure 20: The standard accuracy of AT on natural test data using the *CIFAR-10* dataset with symmetric-flipping and pair-flipping noise. We conduct the experiments using WRN-32-10.

conduct the experiments using the *CIFAR-10* dataset with different noise rates and types. We train the network for 120 epochs and set the weight decay=0.0002, the rest of the settings keep the same as Appendix C.1. We find that AT can still alleviate negative effects of label noise due to memorization effects of deep networks. Particularly, compared with the symmetric-flipping noise, AT has a better performance on avoiding memorization of pair-flipping noise, which can be confirmed by Figures 15 and 16.

## C.4 THE LOSS VALUE

**Result.**    In Figures 21 and 22, we check the loss value of correct/incorrect training data with different noise rates and types. On the whole, compared with ST, correct/incorrect training data can also be more distinguishable in AT using the loss value, regardless of noise rates and types.

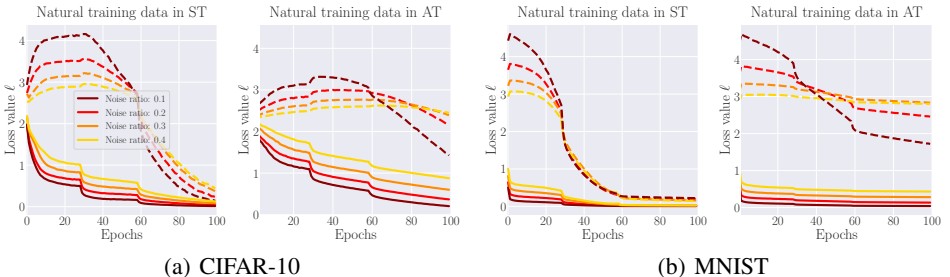

Figure 21: The loss value of ST and AT on correct/incorrect training data using *CIFAR-10* and *MNIST* with *symmetric-flipping* noise. Solid lines denote the loss value of correct training data, while dashed lines correspond to that of incorrect training data. Compared with ST, there is a large gap in the loss value of correct/incorrect training data in AT.

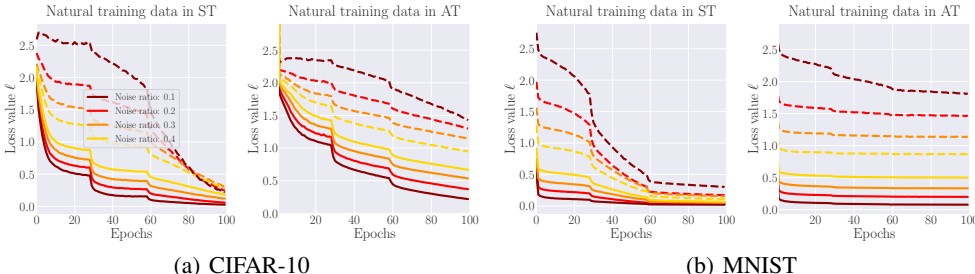

Figure 22: The loss value of ST and AT on correct/incorrect training data using *CIFAR-10* and *MNIST* with *pair-flipping* noise. Solid lines denote the loss value of correct training data, while dashed lines correspond to that of incorrect training data. Compared with ST, there is a large gap in the loss value of correct/incorrect training data in AT.

# D NEW MEASURE: GEOMETRY VALUE $\kappa$

In this section, we provide more experimental results of the geometry value $\kappa$ vs. that of the loss value, and provide more visualization about the specific semantic information corresponds to our new measure. First, we calculate the loss value and geometry value $\kappa$ of correct/incorrect data in AT with different noise rates and types (Appendix D.1). Second, we display more visualization results on *CIFAR-10* and *MNIST* datasets to show the relationship between the geometry value $\kappa$ and image data (Appendix D.2). Moreover, we also discuss the geometry value $\kappa$ with different PGD configurations (Appendix D.3).

## D.1 GEOMETRY VALUE VS. LOSS VALUE

**Result.** In Figures 23 and 24, we plot the density maps of two measures on *CIFAR-10* dataset with symmetric-flipping and pair-flipping noise. We calculate the loss value of natural data and the geometry value in AT using 5 checkpoints at different epochs (e.g., Epoch20, Epoch40, Epoch60, Epoch80, Epoch100), which trained with the same settings in Appendix C.1. We perform the min-max normalization (Tax & Duin, 2000) on both loss value and geometric value $\kappa$, which scales the range of values in $[0, 1]$. On the whole, it is clear that the geometry value $\kappa$ has a stable performance of distinguishing correct/incorrect data under different noise rates and types. Specifically, under the pair-flipping noise with the large noise rate (e.g., Noise rate: 0.4), the loss value cannot differentiate correct/incorrect data well, while the geometry value $\kappa$ can still have a satisfied distinguishing performance.

## D.2 DISTINGUISH RARE AND TYPICAL DATA

**Result.** In Figures 25 and 26, we visualize more results about the semantic information of images corresponding to different $\kappa$ using *CIFAR-10* and *MNIST* datasets. For obtaining the geometry value $\kappa$, we use the GA-PGD method proposed by (Zhang et al., 2021b), which calculates the least number of iterations that PGD needs to find the mis-classified adversarial variants of input data. On *CIFAR-10*, it is calculated by PGD-10 attack with the perturbation bound $\epsilon = 0.031$ and the step size $\alpha = 0.007$. On *MNIST*, the geometry value $\kappa$ is calculated by PGD-40 attack with the perturbation bound $\epsilon = 0.3$ and the step size $\alpha = 0.01$. In general, the geometry value $\kappa$ can represent whether the data is relatively typical or rare.

**About Rare and Incorrect Data.** In AT, the incorrect data are actually far away from the decision boundary, because AT's smoothing effect prevents memorizing the incorrect data (see Figure 1). Therefore, it nearly requires 0 steps. Rare data are near the decision boundary and requires a few steps (e.g., 1 or 2) to find its misclassified adversarial variants.

## D.3 IMPACT OF PGD CONFIGURATIONS

Specifically, we plot more results as Figure 6 with different step size $\alpha$ Figure 27 and the $\epsilon$-ball 28. To be specific, we keep the same setting adopted in Madry et al. (2018) (i.e., step size $\alpha = 2.5 \times \epsilon/n$, where $n$ is the step number). With the same $\epsilon$-ball, we can find that the smaller step size $\alpha$ (i.e., with the larger step number $n$) can provide a nuanced stratification compared with the larger one, which means that the data with similar loss values can have more different $\kappa$ values. With the same step size $\alpha$, a small ball radius $\epsilon$ can not well stratify the data since the PGD attack may never be able to successfully attack some examples.

Back to the proposed geometry value $\kappa$, the key idea behind it is to approximate the distance from a specific sample to the decision boundary. A small step size can provide a nuanced approximation of the distance. Thus, it may be able to provide a nuanced stratification. Since the $\kappa$ value is captured by the adversarial attacks, a large radius $\epsilon$-ball is needed to provide the sufficient attacking strength for a successful adversarial attack to reach the decision boundary. Our previous experimental results have shown that the parameters of PGD in Madry et al. (2018) (widely considered in related literature) is an appropriate choice to realize the correct/incorrect or typical/atypical data stratification in the benchmarked MNIST and CIFAR-10 datasets.

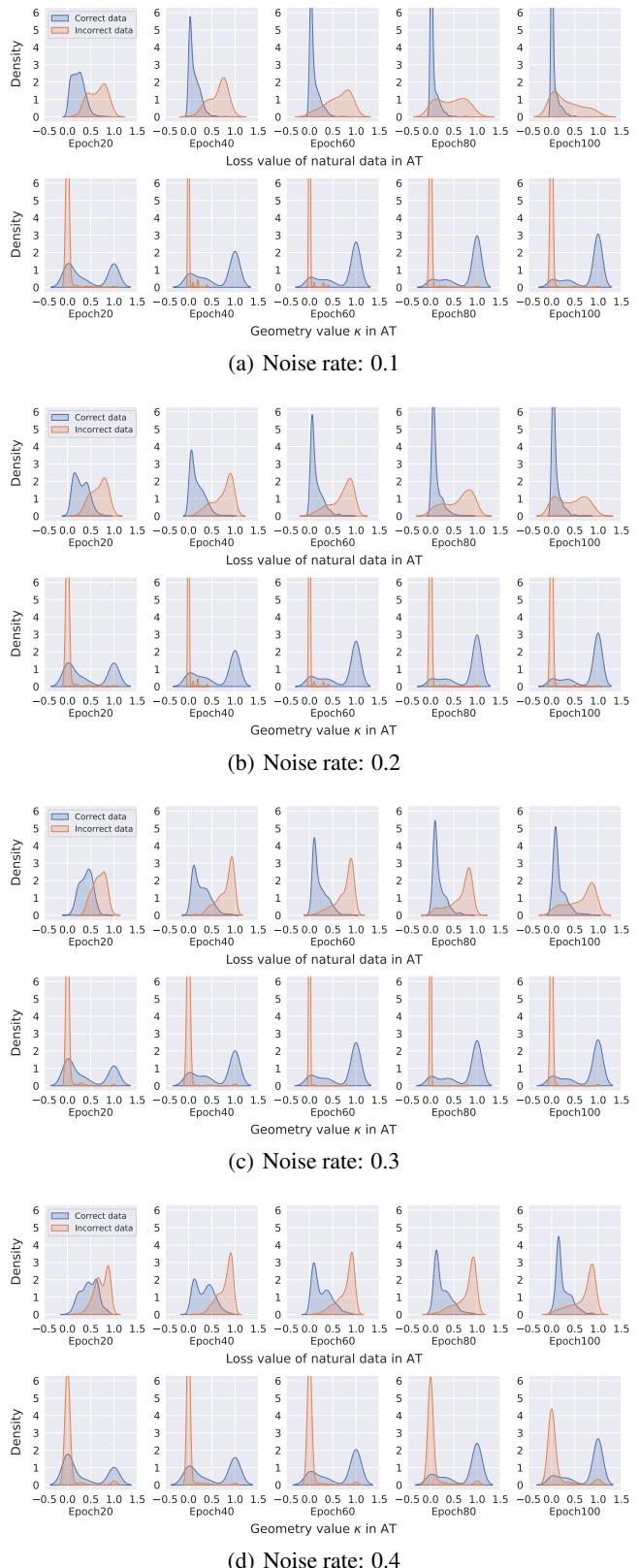

Figure 23: The density of AT on correct/incorrect data using *CIFAR-10* with *symmetric-flipping* noise. *Top panels*: the loss value in AT. *Bottom panels*: the geometry value $\kappa$ in AT. Note that the geometry value $\kappa$ has a better distinction on correct/incorrect data.

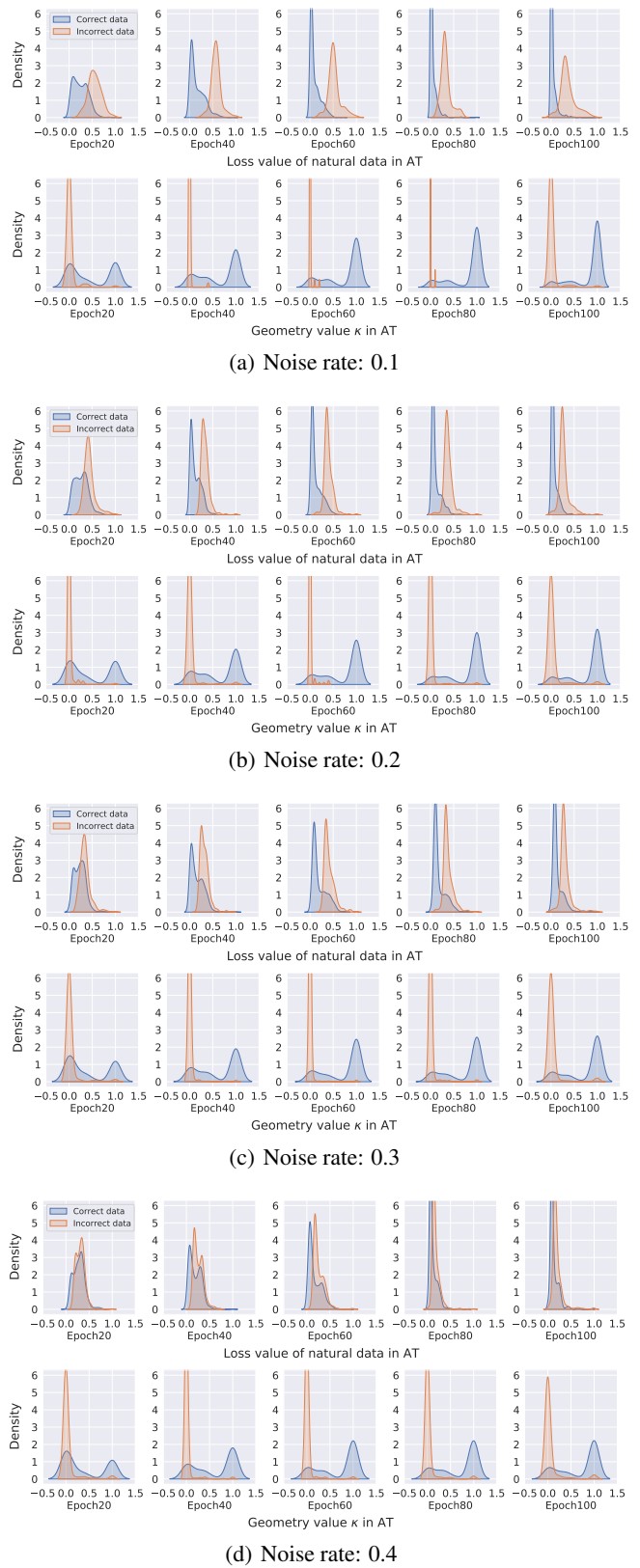

Figure 24: The density of AT on correct/incorrect data using *CIFAR-10* with *pair-flipping* noise. *Top panels*: the loss value in AT. *Bottom panels*: the geometry value $\kappa$ in AT. Note that the geometry value $\kappa$ has a better distinction on correct/incorrect data.

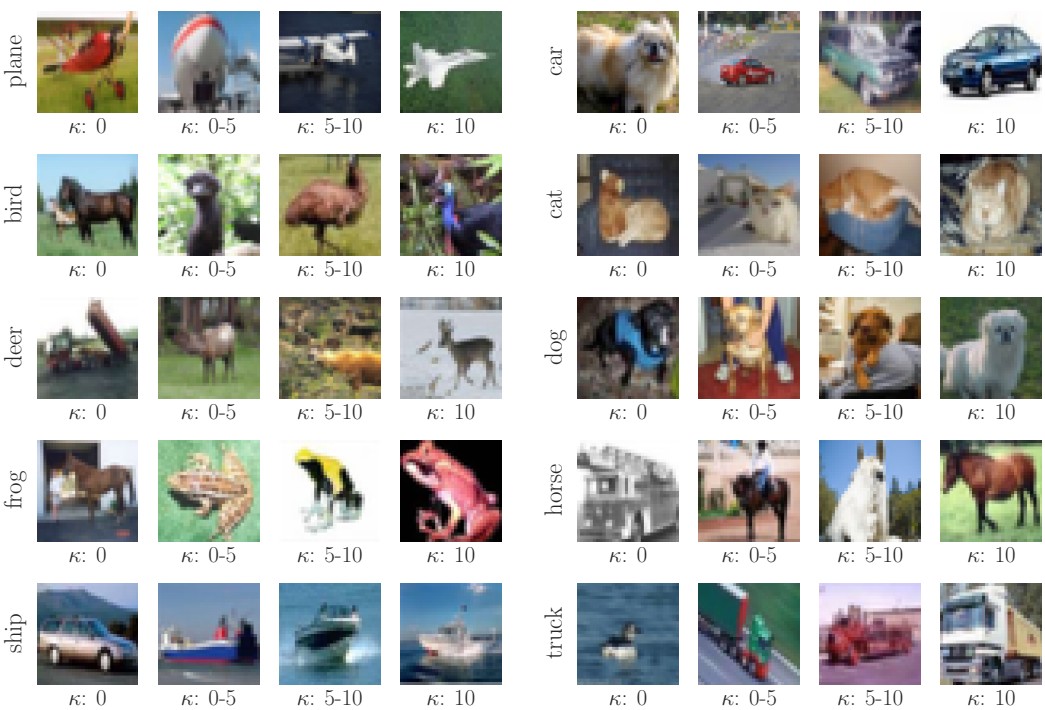

Figure 25: The geometry value $\kappa$ w.r.t. images in *CIFAR-10* with $20\%$ *symmetric-flipping* noise. The leftmost is the given label of all images on the right. We randomly selected four examples with the different $\kappa$ ($\kappa = 0$, $\kappa \in (0, 5)$, $\kappa \in (5, 10)$, $\kappa = 10$) in each class. As the geometric value $\kappa$ increases from left ($\kappa = 0$) to right ($\kappa = 10$), the semantic information of images is more typical and recognizable.

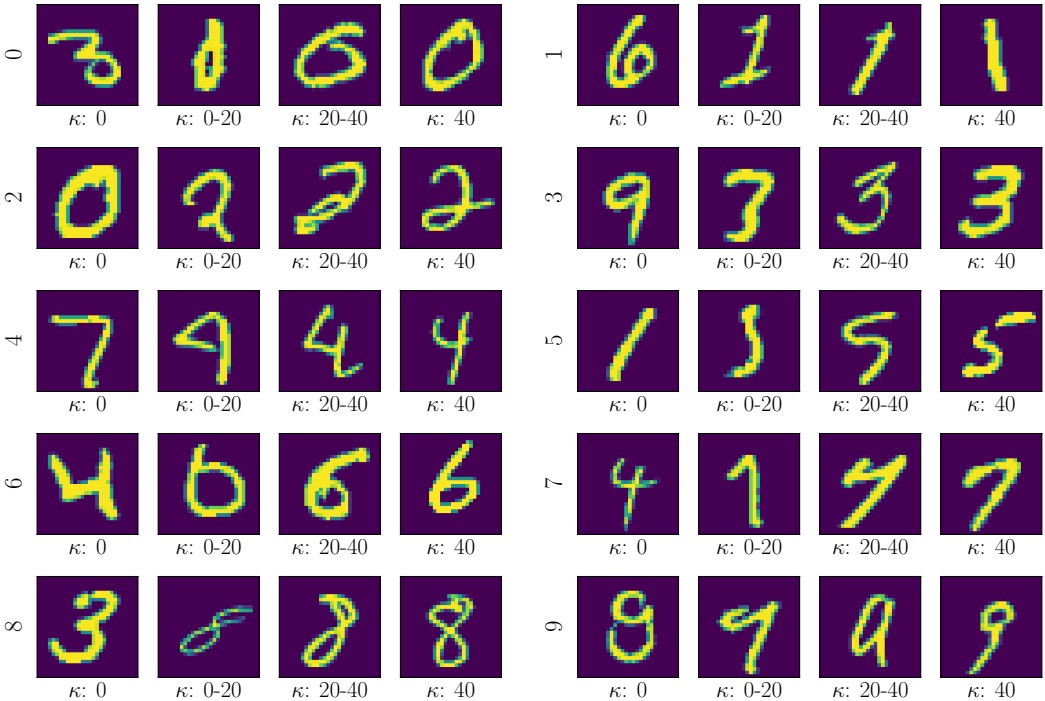

Figure 26: The geometry value $\kappa$ w.r.t. images in *MNIST* with $10\%$ *symmetric-flipping* noise. The leftmost is the given label of all images on the right. We randomly selected four examples with the different $\kappa$ ($\kappa = 0$, $\kappa \in (0, 20)$, $\kappa \in (20, 40)$, $\kappa = 40$) in each class. As the geometric value $\kappa$ increases from left ($\kappa = 0$) to right ($\kappa = 40$), the semantic information of images is more typical and recognizable.

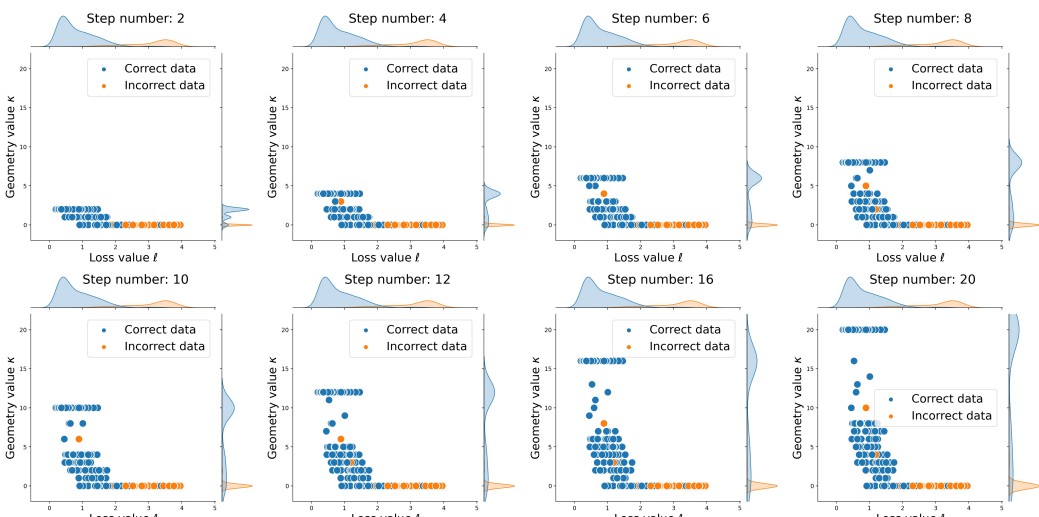

Figure 27: The geometry value $\kappa$ under the different PGD step numbers $n$ with the $\epsilon = 8/255$ and the step size $\alpha = 2.5 \times \epsilon/n$. Note that the smaller step size $\alpha$ (i.e., with the larger step number $n$) can provide a nuanced stratification compared with the larger one.

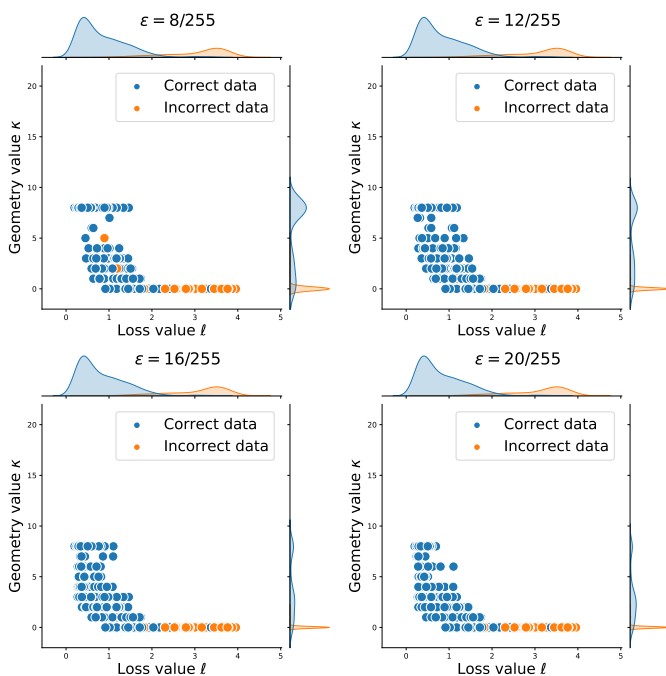

Figure 28: The geometry value $\kappa$ under the different $\epsilon$-ball with the step numbers $n = 8$ and the step size $\alpha = 2.5 \times \epsilon/n$. Note that a small ball radius $\epsilon$ can not well stratify the data since the PGD attack may never be able to successfully attack some examples (i.e., even $n$ steps it can not find a misclassified variant).

# E APPLICATIONS OF GEOMETRY VALUE $\kappa$

In this section, we provide the detailed experimental setups for robust annotator and confidence scores. First, we provide the detailed version of Algorithm 1 (i.e., Algorithm 2) and the details to implement the experiment in Figure 9 (Appendix E.1). Second, we provide the details to implement the experiment in Figure 10 (Appendix E.2).

**Motivation.** Both the two applications are of much significance for obtaining high-quality labeled data. First, since the co-existence of noisy labels and adversarial examples is very realistic, we need to consider training the annotator from noisy training data, and make it robust to adversarial manipulation in unlabeled data. Second, since the label annotations are not always correct, we need the confidence score to show whether the annotation is trustworthy or not.

## E.1 ROBUST ANNOTATOR

In Figure 9, we compare four methods on the *CIFAR-10* dataset, namely, our robust annotator with $20\%$ symmetric-flipping noise, the PGD-based annotator with $20\%$ symmetric-flipping noise, the PGD-based annotator without noise, and the standard annotator without noise.

---

**Algorithm 2:** Robust Annotator Algorithm (in detail).

**Input** : network $f_\theta$, training dataset $S = \{(x_i, y_i)\}_{i=1}^n$, learning rate $\eta$, number of epochs $T$, batch size $m$, number of batches $M$, threshold for geometry value $K$, threshold for loss value $L$.

**Output** : robust annotator $f_\theta$.

**for** epoch $= 1, \ldots, T$ **do**
 **for** mini-batch $= 1, \ldots, M$ **do**
  **Sample:** a mini-batch $\{(x_i, y_i)\}_{i=1}^m$ from $S$.;
  **for** $i = 1, \ldots, m$ *(in parallel)* **do**
   **Calculate:** $\kappa_i$ and $\ell_i$ of $(x_i, y_i)$ by GA-PGD method (Zhang et al., 2021b) and $\ell(f_\theta(x_i), y_i)$, respectively.;
   **if** $\kappa_i < K$ *and* $\ell_i > L$ **then**
    **Update:** $y_i \leftarrow \arg\max_i f_\theta(x)$.;
   **end**
   **Generate:** adversarial data $\tilde{x}_i$ by PGD method (Madry et al., 2018).;
  **end**
  **Update:** $\theta \leftarrow \theta - \eta\nabla_\theta\{\ell(f_\theta(\tilde{x}_i), y_i)\}$.
 **end**
**end**

---

**Choice of Thresholds.** As for the choice of the thresholds in learning with noisy labels, we can use some existing methods to estimate the noise rate of a dataset and then we can set the dynamic thresholds based on the two values of the training data (e.g., greater than or equal to the top 20% largest values).

**Experimental setup.** To generate the noisy training data, we randomly assign the wrong label for a part of correct training data using the method in (Han et al., 2018). For the *CIFAR-10* dataset, we normalize it into $[0, 1]$: Each pixel is scaled by $1/255$. We perform the standard *CIFAR-10* data augmentation: a random $4$ pixel crop followed by a random horizontal flip. For all annotators, we train WRN-32-10 (Zagoruyko & Komodakis, 2016) for 120 epochs using SGD with $0.9$ momentum. The initial learning rate is $0.1$ reduced to $0.01$, $0.001$ and $0.0005$ at epoch $60$, $90$ and $110$. The weight decay is $0.0002$. For standard annotator, we use natural data to update the model. For our robust annotator and the PGD-based annotator, we generate the adversarial data to update the model, the perturbation bound $\epsilon_{\text{train}} = 0.031$, the PGD step is fixed to $10$, and the step size is fixed to $0.007$. All PGD generation have a random start, i.e, the uniformly random perturbation of $[-\epsilon, \epsilon]$ added to the natural data before PGD iterations. For our robust annotator, we use the same generation method with PGD-based annotator as previous mentioned before Epoch 40. After that, we use our Algorithm 2 to

train our robust annotator. We set the threshold for geometry value $K = 2$ and the threshold for loss value $L$ to the loss value of $20\% \cdot m$ largest natural data in each mini-batch, where the $m = 128$ is batch size. We use the model predictions of the selected natural data as their new label to generate the adversarial data. As for the evaluations, we select a part of natural test data on the test set of *CIFAR-10* to add adversarial manipulations by PGD-20 attack. The perturbation bound $\epsilon_{test} = 0.031$, the step number is 20, and the step size $\alpha = \epsilon_{test}/4$, which keeps the same as Wang et al. (2019). We use the natural and adversarial test data to check the performance of annotators on assigning correct labels for the U data. Note that, the experiments are conducted using Tesla V100-SXM2, which takes about 8 hours for each individual trial.

### E.2 CONFIDENCE SCORES

In Figure 10, we plot the accuracy and number of correctly predicted U data w.r.t the geometry value $\kappa$.

**Experimental setup.** We train ResNet-18 model in AT with $20\%$ symmetric-flipping noise on the *CIFAR-10* dataset. The training settings keep the same as Appendix C.1. We use the model checkpoint at Epoch 35 for assigning labels and we randomly select 2000 test data in *CIFAR-10* as unlabeled data. We run the test with 5 repeated times with different random seeds for selecting different test data. In the left panel of Figure 10, we calculate the mean and standard deviation value of accuracy. In the right panel of Figure 10, we show the number of correctly/wrongly predicted data in one of the experiments.

