# OpenReview forum: "Understanding the Interaction of Adversarial Training with Noisy Labels"
_ICLR.cc/2022/Conference — ICLR 2022 Submitted_

### Official Review · Reviewer_FKyS · 2021-10-27

**Correctness:** 3
**Technical Novelty And Significance:** 2
**Empirical Novelty And Significance:** 2
**Recommendation:** 3
**Confidence:** 5

**Main Review:**

This paper studies the usefulness of adversarial training to mitigate the impact of incorrect labelling. I believe this is an interesting topic worth further investigation. However, I have the following concerns regarding the correctness, rigor, novelty, presentation of the paper.

[Part A] Some claims or designs are not rigorous or reasonable.

A1: In the last paragraph of page 4 and the second paragraph of page 5, the authors compare the cases of AT in MNIST and CIFAR10, but I do not believe such comparison is appropriate. I agree that CIFAR10 dataset is more difficult than MNIST, however, we use a much larger adversarial budget on MNIST (0.3) than CIFAR10 (8/255). AT on MNIST is not necessarily easier than CIFAR10.

A2: In Section 3, the authors claim that AT have the smooth effect to prevent the model from fitting the incorrectly labelled data. In this regard, why don't the authors calculate the local Lipchitz constant in the neighbourhood of the incorrectly labelled data? From my point of view, local Lipschitz constant better depicts the smoothness. The Shannon entropy in Equation (3) only demonstrate how confident the model is to the prediction.

A3: In Section 5, the authors claim that the geometry value k is a better metric to distinguish the correct and incorrect labels, while in Algorithm 1, both k and the loss values l are utilised to filter the incorrect labels. This seems contradictory. In addition, it would be difficulty to set an appropriate threshold value L in Algorithm 1. I think the author should conduct ablation study on the condition (the "if" line in Algorithm 1) to filter incorrect labels and show the geometry value is a better metric than loss value.

A4: In Figure 9, the proposed "Robust annotator" have only very marginal improvement when the ratio of adversarial corrupted U data is big. When this ratio is smaller than 0.7, which seems a more reasonable setting, the proposed method is worse than PGD-based annotator.

[Part B] I don't think some discoveries of this paper is novel. Some findings are already included in previous works and is not surprising. Contribution of this paper is limited.

B1: Ref[a] already pointed out ReLU network tends to give overconfident predictions under standard training (ST). In addition, Ref[a] also finds that adversarial training is a kind of calibration to mitigate this overconfident prediction issue. Their findings are very similar, if not identical, to the findings in Section 3 and Figure 2 in this paper.

Ref[a]: Hein, M., Andriushchenko, M., & Bitterwolf, J. (2019). Why relu networks yield high-confidence predictions far away from the training data and how to mitigate the problem. In Proceedings of the IEEE/CVF Conference on Computer Vision and Pattern Recognition (pp. 41-50).

B2: The authors point out AT can better preserve the standard test accuracy than ST when the training data has label noise. This finding is of little practical usage. For example, ST can utilise a label noise-free validation set, which is much smaller than the training set, to do early stopping. Through this way, ST can quickly achieve nice clean test accuracy and is faster than AT.

B3: The contributions of this paper is limited on top of Ref[b]. Ref[b] theoretically and empirically demonstrate that networks that converge to data of label noise have high adversarial vulnerability. The equivalent claim of this is network that is adversarially robust should have larger loss on the training data of label noise. This is exactly that Figure 1 and Section 3 in this paper discuss. Ref[b] has theoretical justifications while this paper only provides empirical evidence.

Ref[b]: Sanyal, A., Dokania, P. K., Kanade, V., & Torr, P. (2020, September). How Benign is Benign Overfitting?. In International Conference on Learning Representations.

B4: This paper lacks theoretical justifications and quantitative study on this phenomenon in terms of the perturbation bound $\epsilon$. We know ST is a special case of AT with $\epsilon = 0$. With the increase of $\epsilon$, ST can gradually shift to AT. I believe adversarial training with small values of $\epsilon$ would demonstrate a more similar behaviour to ST. A quantitative analysis on this needs to be conducted.

[Part C] The presentation and writing of some part need improvement.

C1: I think Equation (3) is the definition of Shannon entropy, why it is $p(x, y) log p(y|x)$ instead of $p(y|x) log p(y| x)$. Note that $\sum_y p(y|x) = 1$ but $\sum_y p(x, y) \neq 1$. We do not know $p(x, y)$ or estimate it in practice, because $p(x, y)$ is the joint distribution.

C2: Some wordings need improvement. For example, in the first point of the contribution: "AT can always distinguish ..." I do not think AT can ALWAYS does so. There are several similar mistakes in other sections.

**Summary Of The Paper:**

This paper studies the adversarial training in the context of label noises. Specifically, it is discovered that adversarial training can prevent the model from overfitting to the label noises, leading to a more smooth landscape. In addition, the authors point out that the number of PGD step sizes can be considered as a useful metric to distinguish instances with correct or incorrect labels.

**Summary Of The Review:**

Based on the questions and concerns above. Unfortunately, I cannot recommend acceptance of this manuscript to ICLR. I welcome the author to post rebuttals and will do a re-evaluation after the our discussion ends.

---

### Official Review · Reviewer_sbre · 2021-10-30

**Correctness:** 2
**Technical Novelty And Significance:** 2
**Empirical Novelty And Significance:** 2
**Recommendation:** 5
**Confidence:** 4

**Main Review:**

**Strengths:** The use of _PGD step number_ is intuitive and this intuition is verbally justified in my opinion. Detailed analysis is conducted on CIFAR-10 to empirically justify the benefit of _PGD step number_ over the loss value. The _robust annotator_ method proposed by the authors seems to be bringing improvement to the dataset in consideration (CIFAR-10).

**Weaknesses:** The main weakness of this paper is the lack of sufficient evidence to support the claims and the use of _PGD step number_.  In detail,

* The claims made in the paper are often overgeneralizing, see [1] for some examples. **Question:** I have difficulty understanding how these claims are made so confidently. What evidence are they based on? I understand the intuition from the decision boundary perspective, but I do not think only CIFAR-10 experiments in a very specific setting (pair-flipping noise) are enough to support all these claims. Can the authors clarify?

* Similarly as above, the justification is made neither through a rigorous analysis nor an extensive set of experiments. It does not have to be a theoretical analysis, but only CIFAR-10 experiments with the robust annotator and confidence scores are very limited. **Question:** when will the use of PGD step number break? what are the limitations? what happens for instance when the data has a large number of classes or in more extended noise settings? There is some mention of it in the appendix (CIFAR-100, asymmetric-flipping), but it is either not sufficient enough to get an answer to this in my opinion (CIFAR-100) or must be moved to the main body as needing to read 27 pages (3x page limit) should be prevented in order to see sufficient amount of justification. I think more than half of the main body is giving background; the paper could be written such that the contributions are more richened and highlighted.

[1]  Main contributions on pg. 2, the first paragraph of Section 4


**Note to the authors:** I'm willing to review my assessment based on the authors' answers.

**Summary Of The Paper:**

This paper focuses on understanding adversarial training in the presence of label noise by conducting empirical studies. Based on their observations, the authors propose to use _PGD step number_ of adversarial training as a new measure for sample selection to correct noisy labels. Moreover, they present two use-cases, namely 1. a _robust annotator_ algorithm to label unlabeled instances, and 2. PGD step number as a _confidence score_ for the labeling of unlabeled instances. Empirical observations are primarily made on CIFAR-10 images.

**Summary Of The Review:**

Although the use of _PGD step number_ seems promising, there is only very limited evidence to support the claims in the paper. That is, there is neither a rigorous analysis/formulation nor a comprehensive empirical study to justify the use of _PGD step number_. Hence, I believe significant improvement is needed for this paper to back up its claims, leading my decision to a rejection.

---

### Official Review · Reviewer_BbnD · 2021-11-02

**Correctness:** 3
**Technical Novelty And Significance:** 3
**Empirical Novelty And Significance:** 3
**Recommendation:** 6
**Confidence:** 3

**Main Review:**

> Strength of this paper: this paper addresses an interesting and novel problem connecting adversarial training and noisy labels. The writing and clear and easy to follow. The proposed method is intuitive yet effective. Extensive experimental studies were conducted.

> Main concern: The results are highly dependent on the hyper-parameter values, for example, epsilon (adversarial attack budget), and learning rate alpha in the PGD attack. Intuitively, lower epsilon and lower alpha can both lead to a higher number of iteration in the PGD attack.

> In Figure 7, the overlap between correct and incorrect data's density of AT is large. I am wondering if there is a confidence value or statistics number to measure the distributional difference? I am not sure if merely mean/std can capture this difference.


**Summary Of The Paper:**

This paper studies the connection between noisy labels (NL) and adversarial training (AT). The contribution of this paper is two-fold. The first one is to adopt the number of PGD attack steps as a criterion for sample selection to correct noisy labels. The second one is that adversarial training can serve as a way to correct noisy labels. These two contributions indicate that adversarial training can be applied to more general model robustness problems.



**Summary Of The Review:**

An interesting connection is studied and the contribution is clear. But the work lacks theoretical insight and quantitative results. So I would recommend only a borderline accept.

---

### Official Review · Reviewer_cfM4 · 2021-11-03

**Correctness:** 3
**Technical Novelty And Significance:** 2
**Empirical Novelty And Significance:** 2
**Recommendation:** 5
**Confidence:** 3

**Main Review:**

Strengths:

- The question they study is certainly well-motivated: label noise and adversarial robustness appear to touch on two separate notions of robustness, yet it seems plausible that adversarial training might avoid running into the issues that vanilla training has with overfitting noisy labels in the training data. This paper explores this potential connection from a variety of different angles beyond just the basic question of whether AT can improve generalization in the presence of label noise.
- The "geometry value" seems like a useful proxy for discerning incorrectly labeled points from correctly labeled points.

Weaknesses:
- As the paper mentions, the general question of whether AT can avoid memorizing noisy labels has been studied in previous work, e.g. [Sanyal et al. 2021], and the geometry value was introduced in a previous work [Zhang et al. 2021b] as a proxy for the distance to the decision boundary. My understanding is that on the former front, the present paper mainly differs from [Sanyal et al. 2021] in that it provides a candidate explanation for this phenomenon via the so-called ``smoothing effect.'' I found the description of the "smoothing effect" vague, but more importantly, I wasn't very convinced that the experiments in the smoothing effect section were sufficiently meaningful. Certainly the 2D synthetic example is too limited in scope to draw major conclusions. I was also confused by the average entropy experiment because it doesn't seem to be related to what the 2D experiment in Figure 1 is suggesting. E.g. take the top-right incorrectly labeled point in Figure 1. In the left plot, the distribution over labels in a neighborhood of that point is concentrated around yellow (corresponding to red), and in the right plot, it's concentrated around black (corresponding to blue). So the entropies in the two cases appear to be comparable? I might just be missing something basic, but it would be helpful at least for me to have more discussion in Section 3 clarifying this point. A related confusion I had about the meaning of the "smoothing effect" is that [Sanyal et al. 2021] observed that "adversarial training does produce more complex decision boundaries," whereas the qualitative picture I get out of Section 3 seems to be the opposite?
- While the geometry value experiments in Section 5 were interesting, it wasn't clear from Figure 10 that it's a useful proxy for annotation confidence. There, because there are so few points with geometry value between 6 and 9, the lower end of the confidence intervals in the left plot in Figure 10 are essentially the same for values 4 to 9.
- A minor comment: while the abstract makes clear that geometry value was introduced in prior work, various parts of the intro (e.g. P. 2) give off the impression that the definition of geometry value is novel to this submission, so perhaps the wording in these places could be fixed?

**Summary Of The Paper:**

This submission empirically studies the efficacy of adversarial training for mitigating the effect of label noise in training data. Their findings are as follows:
1) "Smoothing effect" of adversarial training:
	a) on a 2-dimensional synthetic binary classification dataset where two points are incorrectly labeled, they show that vanilla training yields a classifier that memorizes the bad labels by forming "clusters" around the incorrectly labeled points, whereas adversarial training does not yield such clusters
	b) for CIFAR injected with 20/40% random label noise, they ran vanilla and adversarial training on the noisy data and found that if you look at the distribution over labels within the neighborhood of a random incorrectly labeled point, on average the entropy of that distribution is higher for the classifier obtained by adversarial training than by vanilla training
2) For vanilla training on CIFAR (also MNIST) injected with label noise, the gap between accuracy on the correctly labeled training data and the incorrectly labeled data closed over the course of training, whereas this gap does not close or seems to close much more slowly for adversarial training.
3) Over these same noisy datasets, adversarial training seems to mitigate the impact of noisy training data on (clean) test accuracy, unlike vanilla training for which generalization degrades as label noise increases
4) They consider a quantity they call the "geometry value" of a data point (x,y), which corresponds to the number of PGD steps needed to find a differently labeled point in the neighborhood of x. This quantity was originally introduced by [Zhang et al. 2021b], and in the present paper they find that:
	a) compared to loss(x,y), it appears to be a more effective way to effective way to distinguish correctly labeled data from incorrectly labeled training data, as well rare data from typical data
5) They propose a "robust annotator" for labeling unlabeled data that has possibly been subject to adversarial perturbations. The algorithm repeatedly alternates between 1) identifying training data points with high loss and low geometry value and re-labeling them according to the current classifier and 2) running a step of adversarial training. It appears to do slightly better than a PGD-based annotator baseline when trained over CIFAR injected with label noise. They also note that the geometry value can provide some kind of "confidence score" to go along with the label annotations.

**Summary Of The Review:**

The experiments on geometry value were interesting, and the paper asks an important question in trying to relate label noise and adversarial robustness. That said, for the results on adversarial training mitigating overfitting of noisy labels, I'm not sure I fully grasp what the key new conceptual insights are relative to prior work as I wasn't particularly convinced by the definition of or the experiments on the "smoothing effect" of adversarial training. So I'm a bit on the fence and leaning towards weak reject, though I'm happy to update my score if the authors could help clarify my confusions above.

---

### Decision · Program_Chairs · 2022-01-20

**Decision:**

Reject

**Comment:**

All reviewers agree that the proposed idea looks interesting but the paper is seriously lacking in the definition of its scope: there is no quantitative result, experiments are quite limited, and there is not enough discussion of the limitations. With more work this could become a very interesting paper.